# Acute Appendicitis as the Initial Presentation of Kawasaki Disease Shock Syndrome in Children

**DOI:** 10.3390/children9121819

**Published:** 2022-11-25

**Authors:** Yuan-Hao Chang, Chien-Yu Lin, Lu-Hang Liu, Fu-Huan Huang, Yu-Jyun Cheng

**Affiliations:** 1Department of Pediatrics, Hsinchu MacKay Memorial Hospital, Hsinchu City 30071, Taiwan; 2Division of Pediatric Surgery, Department of Surgery, Taipei Medical University Hospital, Taipei City 110, Taiwan; 3Department of Pediatrics, Changhua Christian Children’s Hospital, Changhua City 50050, Taiwan

**Keywords:** Kawasaki disease, Kawasaki disease shock syndrome, appendicitis, intravenous immunoglobulin, coronary artery dilatation

## Abstract

Kawasaki disease shock syndrome (KDSS) is a severe form of Kawasaki disease (KD). The hemodynamic instability and atypical manifestations of this syndrome delay its correct diagnosis and timely treatment. We report here an eight-year-old girl who presented with appendicitis. Her fever persisted after appendectomy, accompanied by hemodynamic instability. The girl was diagnosed with KDSS. Intravenous immunoglobulin (IVIG) and corticosteroids were administered. Her symptoms resolved. She had left coronary artery dilatation, which resolved three months later. We also reviewed two other possible cases identified as KDSS with appendicitis. These cases have a more atypical clinical course, prolonged treatment, and a higher rate of IVIG resistance. Better awareness of KDSS is needed for early diagnosis and treatment in children experiencing prolonged fever after appendectomy.

## 1. Introduction

Kawasaki disease shock syndrome (KDSS) is a severe form of Kawasaki disease (KD) [1]. KD, as one of the more common forms of vasculitis in young children, is characterized by the following features: unexplained prolonged fever lasting more than 5 days, skin rashes, bilateral bulbar conjunctival injection, erythema of the lips and oral mucosa, non-suppurative cervical lymphadenopathy, and changes in peripheral extremities [2,3]. In recent years, a subgroup of children with Kawasaki disease developing hemodynamic instability has been further studied. Kanegaye et al. define this as a rare manifestation of KD characterized by systolic hypotension or clinical signs of poor perfusion that require volume expansion and infusion of vasoactive agents [4]. These patients have severe markers of inflammation, a higher rate of coronary abnormalities, and intravenous immunoglobulin (IVIG) resistance [5,6]. Their initial presentation is also not as typical as KD. Acute appendicitis is one of the rare initial leading manifestations. Huang et al. report serial cases of appendicitis in Kawasaki disease [7]. None of the cases showed KDSS.

## 2. Case Report

A previously healthy eight-year-old girl presented with fever for 1 day, accompanied by several days of vomiting and non-bloody, watery diarrhea. She also suffered from periumbilical abdominal pain for 5 days. The location of the pain shifted to the right lower quadrant 1 day before her arrival. She was brought to our outpatient department for suspicion of acute appendicitis. She had no cough, conjunctivitis, rhinorrhea, joint pain, or consciousness disturbance. A plain radiograph showed diffuse bowel ileus. The number of leukocytes in the blood was 19,800/μL with neutrophiles predominant (88%), and the level of C-reactive protein (CRP) was 30.67 mg/dL. In addition, prolonged INR (1.45) and APTT (38.7 s) and elevated D-dimer (7247.00 ng/mL) and erythrocyte sedimentation rate (ESR) (67 mm/h) were also noticed. Her platelet count was 337,000/μL. Her blood pressure was 115/74 mmHg, and her heart rate was 95 beats per minute. Acute appendicitis, leading to sepsis and disseminated intravascular coagulation, was initially suspected. Empirical antibiotics were prescribed along with intravenous fluid resuscitation. Computer tomography scan of the whole abdomen revealed borderline wall thickening of the appendix. She received a laparoscopic appendectomy on the second day of admission due to persistent abdominal pain with peritoneal signs. However, after the operation, she presented with hypotension, tachycardia, and oliguria. No signs of bleeding were observed. She was transferred to the intensive care unit. Broad-spectrum antibiotics and inotropic agents were administered. Three days after surgery, a bilateral bulbar non-exudative conjunctival injection developed. Swelling and erythematous change in digits were also noted (Figure 1). Echocardiography showed a decrease in systolic function with the use of dopamine and dobutamine, and the ejection fraction showed only 54.4%. Pericardial effusion and mitral regurgitation were also noticed. There was no coronary artery dilatation initially. Elevated troponin-I (0.91 ng/mL) was also found. Elevated white blood cell counts (20,500/μL), ESR (38 mm/h), and CRP (20.85 mg/dL) were notable in repeated blood tests. No other viral or bacterial pathogen was isolated. The diagnosis of Kawasaki disease shock syndrome was highly suspected. Therefore, we prescribed aspirin (4 mg/kg/day) and IVIG (1 g/kg/day). Her fever soon subsided. She was weaned from inotropic agents. She was transferred to the ordinary ward on the ninth day after admission. Pathologic reports of her appendix showed infiltration of acute inflammatory cells in the appendix (Figure 2), with no mucosal erosion–ulceration (Figure 3). The picture of the appendix was not compatible with the clinical finding of turbid ascites. A concomitant inflammatory source in another area was suspected. She was discharged on the thirteenth day. However, she developed a fever again 20 days after the onset of the disease. She also developed muscle pain and neck pain. The swelling and pain in her feet improved, but red eyes were again noted. Red lips and strawberry tongue were also observed. There were no skin rashes. Elevated white blood cell counts (10,600/μL), ESR (109 mm/h), CRP (13.58 mg/dL), and D-dimer (3133.00 ng/mL) levels were noted. A Group A streptococcus rapid test showed positive. She was admitted again and given an antibiotics treatment. Echocardiography was performed. Coronary dilatation of the left coronary artery (LCA) (0.48 cm; Z-score = 3.15) and right coronary artery (RCA) (0.252 cm; Z-score = 0.15) were found (Figure 4). Some maculopapular rashes developed on four limbs 3 days later, and fever persisted. Refractory KD was diagnosed. Immunoglobulin (2 g/kg/day), high-dose aspirin (47.2 mg/kg/day), and oral prednisolone (60 mg/day) were administered. Her fever subsided 2 days later after IVIG administration. We changed the aspirin to low dose (4 mg/kg/day) and kept it for 5 months. Skin rashes subsided 3 days later after IVIG use. Desquamations of the fingers were later found. Red eyes and lips also improved. Prednisolone gradually tapered after 6 weeks of treatment. After 3 months of follow-up, the coronary dilatation subsided. The child recovered completely and was regularly followed up in the outpatient department.

## 3. Discussion

The percentage of possible KDSS in children ranges from 1.5% to 7% [8,9,10]. In a recent study, it was found that KDSS in patients is underreported, and more aggressive treatment is needed [11]. Furthermore, more severe gastrointestinal symptoms are observed in children with KDSS [1,6].

KD may present with appendicitis, though it is a rare presentation. There is little information about KDSS with appendicitis. Two possible cases are reported after reviewing the literature (Table 1). Taddio, A. et al. reported a patient with KDSS undergoing a surgical intervention due to a clinical and radiological diagnosis of appendicitis [9]. However, post-operative findings showed thickening of the ileal loops, consistent with mesenteric vasculitis. Gamez-Gonzalez, L.B. et al. reported another possible case [1]. One patient underwent a surgical intervention as a result of a clinical and radiological diagnosis of appendicitis and septic shock. The postoperative findings were positive only for peritoneal cavity effusion. In our case, the clinical diagnosis of appendicitis was made. The pathology report of the appendix showed the presence of neutrophils in the muscularis propria of the appendiceal wall, which was compatible with appendicitis. However, the mucosa of the appendix was intact, which may imply that the inflammatory source was not the appendix.

Appendicitis as a manifestation of KD occurs mainly in children between 3 and 7 years of age [7]. Our case is an eight-year-old girl. Patients with surgical-onset KD were older than patients with nonsurgical-onset KD, and most of them were older than 5 years old [12]. Among these children, complete KD occurs in 77% (10/13) of patients [7,13]. Of the three cases of KDSS with appendicitis, there was one incomplete KD [1,9]. This may be related to a more severe inflammatory process and atypical presentation in patients with KDSS. We can also find that 77% (10/13) of KD patients with appendicitis have a diagnosis with histologic reports [7,13]. Our case had the pathologic confirmation while the remaining cases with KDSS did not [1,9].

In children of KD presenting with appendicitis, 30% (4/13) have cardiac involvement with coronary dilation or aneurysms [7,13]. Children with KDSS seem to have a higher rate of coronary artery abnormalities than children with KD [8,9,10]. Furthermore, left ventricle dysfunction represented by reduced ejection fraction (<55%) were found to be more common in the KDSS group than in the hemodynamically stable KD group in children [6,14]. The more severe inflammatory process with increased cardiac involvement may lead to hemodynamic instability in children with KDSS.

Furthermore, children with KDSS have a higher IVIG resistance rate [10,11,14,15]. Significantly increased serum interleukin 6 (IL-6) in the acute phase and increased vascular endothelial growth factor in the late phase [16] are observed in patients with KD who are resistant to initial IVIG treatment. Higher serum interleukin 10 (IL-10) levels were also found in patients not responding to IVIG [17]. Serum procalcitonin elevation was observed in the KDSS. Increased serum procalcitonin level had been reported to be associated with more IVIG resistance and more coronary artery dilatations. Elevated serum procalcitonin in patients with KD may be a useful biomarker of multiple organ damage, predicting the KDSS [17,18]. Hyperactivation of the interferon-γ and TNF-α signaling pathways were also mentioned [19]. The amount of anti-cytokine antibodies contained in IVIG may be insufficient to block excessive cytokines, although other mechanisms are yet to be investigated [17]. Therefore, early administration of corticosteroids with IVIG may be considered in these patients to avoid treatment failure and prevent coronary abnormalities [20]. However, some severe adverse events in the patient who received IVIG plus corticosteroids treatment were reported [21], including hypotension, respiratory dysfunction, neutropenia, and hypercholesterolemia. Careful monitoring of these side effects during the treatment course is also important.

This is the first formal report of acute appendicitis as the initial manifestation of KDSS. We should be aware of KD when the patient demonstrates persistent fever and other signs of KD after the appendectomy, especially when no other possible infectious sources are identified. Furthermore, hemodynamic instability warranted a more severe inflammatory process and more possible treatment failure. More aggressive treatment strategies upon the diagnosis of KD and KDSS should be considered.

The patient was admitted in March 2018. At that time, coronavirus disease 2019 (COVID-19) was not reported.

## 4. Conclusions

Appendicitis is a rare presentation of KD, and it also occurs in patients with KDSS. This unusual presentation is more common in older children. These patients have a more atypical clinical course and prolonged treatment and hospital stays. If persistent fever and hemodynamic instability with some clinical features of KD are noted after the appendectomy, KDSS should be considered. Early recognition of the disease may allow for the combined use of IVIG and corticosteroids, which can reduce coronary artery abnormalities and improve treatment outcome.

## Figures and Tables

**Figure 1 children-09-01819-f001:**
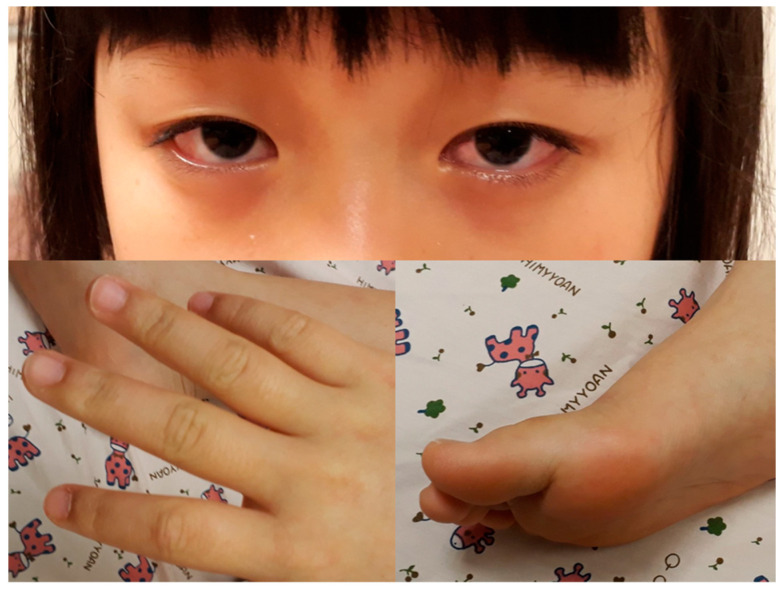
Bilateral bulbar non-exudative conjunctival injection, swelling, and erythematous change of fingers and soles.

**Figure 2 children-09-01819-f002:**
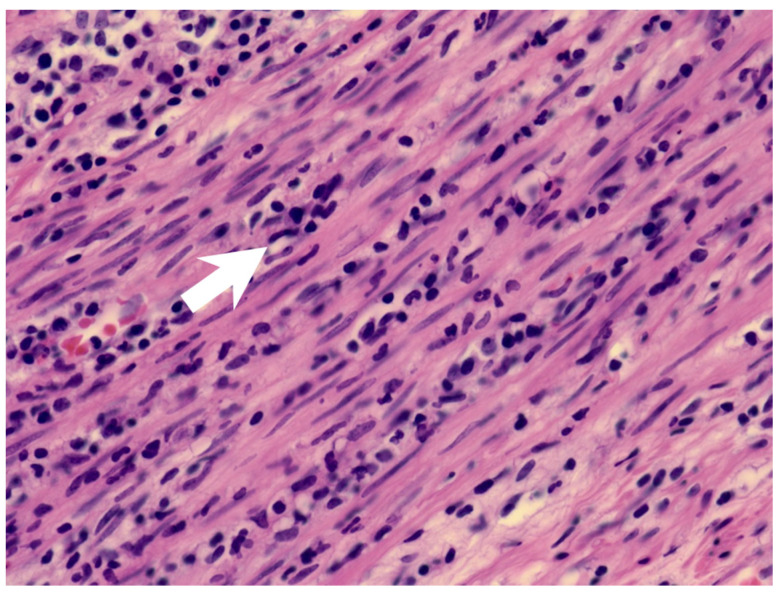
The muscularis layer of the appendix shows infiltration of acute inflammatory cells (arrow).

**Figure 3 children-09-01819-f003:**
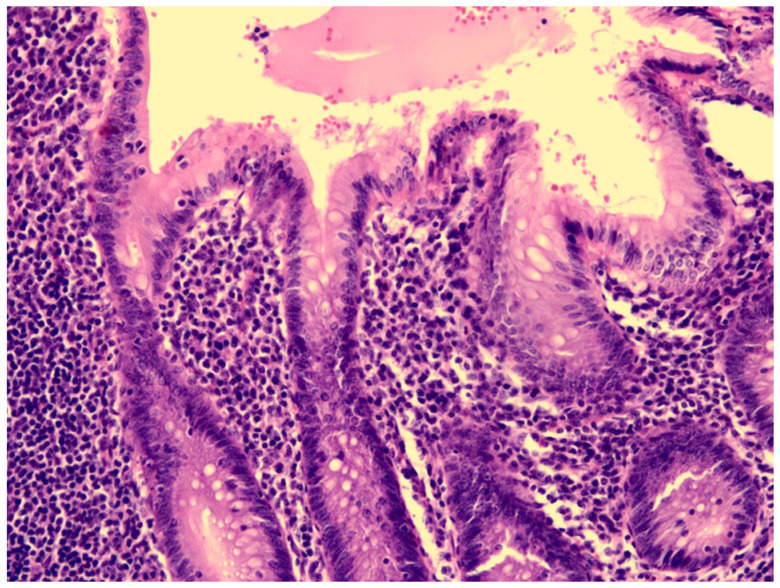
The mucosa of the appendix was intact, without erosion or ulceration.

**Figure 4 children-09-01819-f004:**
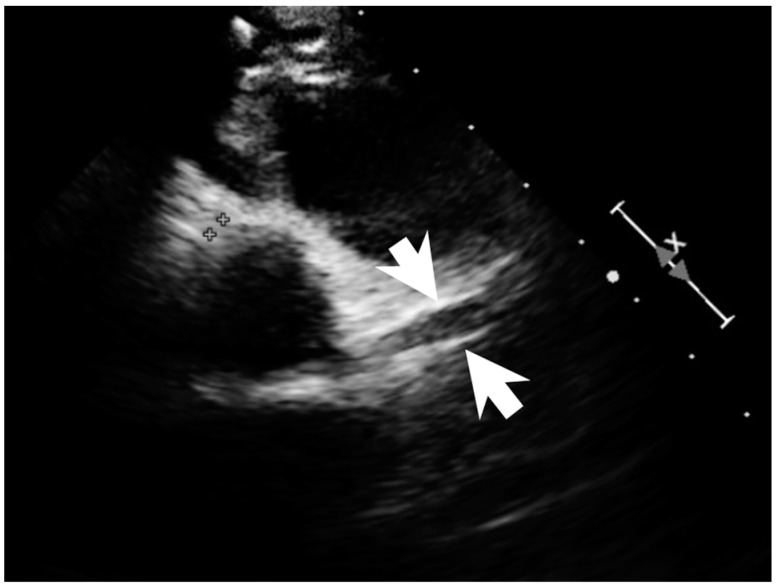
Coronary dilatation of LCA on the echocardiography (arrows).

**Table 1 children-09-01819-t001:** Clinical characteristics of Kawasaki Disease Shock Syndrome with appendicitis in 3 children.

Reference Study (Publication Year)	Surgical Intervention	Postoperative Findings	Pathological Confirmation of Appendicitis	KD Type
Our patient	Yes	Hyperemic and engorged appendix with fibrin coating	Yes	Complete KD
Taddio, A. et al. (2017) [9]	Yes	Thickening of the ileal loops, consistent with mesenteric vasculitis	No	NA
Gamez-Gonzalez, L.B. et al. (2013) [1]	Yes	Positive for peritoneal cavity effusion	No	Incomplete KD

Abbreviations: KD, Kawasaki disease; NA, not available.

## Data Availability

Not applicable.

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
