# Peer review of "Acute Appendicitis as the Initial Presentation of Kawasaki Disease Shock Syndrome in Children"

_children, 2022, doi:10.3390/children9121819_

Round 1

Reviewer 1 Report

I have no comments

Author Response

Thank you very much for your careful review. We appreciate the time you spend to help our article better.

Reviewer 2 Report

The case report "Acute appendicitis as the initial presentation of Kawasaki disease shock syndrome in children" describes an 8-year-old child with a diagnosis of KDSS following appendectomy. Details of the case, including illness presentation, progression, and treatment, as well as commentary regarding its rarity and significance, are appropriately provided. The report would benefit from further English language editing; there are minor errors throughout.

Author Response

Thank you very much for your careful review and suggestion. We have completed the English language editing and we provide the certification of English editing in the attachment. 

Reviewer 3 Report

Dear authors,

Thank you for submitting your manuscript to Children. I would like to congratulate for your good work on this case report and review of the literature and for your effort to ensure readers a better view of this atypical form of presentation of Kawasaki disease. 

A few minor suggestion should be considered: having so little data on reported cases, it might be advisable to create a table with all the cases found and their characteristics. IAlso, in order to have a better view of the clinical course of the patient it is recommended to mention the exact medical treatment, including doses and step-up/step-down management.
